# Research on Evaluating the Sustainable Operation of Rail Transit System Based on QFD and Fuzzy Clustering

**DOI:** 10.3390/e22070750

**Published:** 2020-07-07

**Authors:** Bing Yan, Liying Yu, Jing Wang

**Affiliations:** 1School of Urban Railway Transportation, Shanghai University of Engineering Science, Shanghai 201620, China; yanbinged@163.com; 2School of Management, Shanghai University, Shanghai 200444, China; yuliying@shu.edu.cn

**Keywords:** rail transit system, sustainable operation, PSR, QFD, fuzzy clustering

## Abstract

The purpose of this study is to evaluate the sustainable operation of rail transit system. In rail transit system, as the most important aspect of negative entropy flow, the effective strategy can offset the increasing entropy of the system and make it have the characteristics of dissipative structure, so as to realize the sustainable operation. At first, this study constructs the Pressure-State-Response (PSR) model to evaluate the sustainable operation of rail transit system. In this PSR model, “pressure” is viewed as customer requirements, which answers the reasons for such changes in rail transit system; “state” refers to the state and environment of system activities, which can be described as the challenges of coping with system pressure; “response” describes the system’s actions to address the challenges posed by customer needs, namely operational strategies. Moreover, then, 13 pressure indices, five state indices and 11 response indices are summarized. In addition, based on quality function deployment (QFD), with 13 pressure indices as input variables, five state indices as customer requirements (CRs) of QFD and 11 response indices as technical attributes (TAs) of QFD, this study proposed the three-phase evaluation method of the sustainable operation of rail transit system to obtain the operational strategy (that is, negative entropy flow): The first phase is to verify that 13 pressure indices can be clustered into five state indices by fuzzy clustering analysis; The second phase is to get the weights of five state indices by evidential reasoning; The third phase is to rate the importance of 11 response indices by integrating fuzzy weighted average and expected value operator. Finally, the proposed model and method of evaluation are applied to the empirical analysis of Shanghai rail transit system. Finally, we come to the conclusion that Shanghai rail transit system should take priority from the following five aspects: “advancement of design standards”, “reliability of subway facilities”, “completeness of operational rules”, “standardization of management operation” and “rationality of passenger flow control”.

## 1. Introduction

The construction of the subway has improved the public transportation system of cities, which not only easing traffic jams, but also developing the quality of people’s life. An integrated, efficient and economical rail transit system, which is an important condition for sustainable development of urbanization. However, in order to get the benefits of the subway, the operation strategy of rail transit system must be implemented [1].

For very large cities, the shortage of transportation capacity during peak hours and the relative shortage of passengers during off-peak hours, which pose a challenge to the operation of rail transit [2]. In addition, during the rush hour, safety issues are also a huge challenge for the subway operation [3,4]. Some researchers have focused on safety issues about the sustainable operation of rail transit [5,6,7,8]. Additionally, some scholars also focus on energy conservation strategies and service quality levels on sustainable operation of rail transit [9,10,11,12,13]. Recently, customer demand is particularly important in the study of sustainable operation of rail transit [14,15]. Following the people-oriented principle, it is necessary to consider sustainable operation from the perspective of customers [15]. Therefore, this study aims to achieve the sustainable development from the perspective of customer requirements, so as to correctly solve the following problems: (1) How to describe the relationship between customer requirements and operational strategies by establishing a sustainable and operational rail transit system? (2) How to transform customer requirements into sustainable operational strategies? (3) How to analyze the characteristics of customer requirements and quality them? (4) How to determine the weights of customer requirements and the importance ranting of operational strategies? These are of great significance to improve the safety and service quality of rail transit and further realize sustainable operation.

The structure of this study is as follows: In Section 2, literature review is carried out from the perspective of sustainable operation of rail transit system and related methods. In Section 3, this study shows definitions of Dempster–Shafer evidence theory and fuzzy set theory, respectively. In Section 4, the sustainable operation of rail transit system is introduced first. Then, the evaluation model and indices of sustainable operation of rail transit system based on PSR are constructed. Finally, based on QFD, this study develops the three-phase evaluation method of sustainable operation of rail transit system. In Section 5, an empirical analysis on the sustainable operation of Shentong Metro is made. In Section 6, conclusions and sustainable recommendations are provides.

## 2. Literature Review

### 2.1. The Sustainable Operation of Rail Transit System

Rail transit system is a complex system, which integrates multiple professions and different types of work. It is usually composed of rail routes, stations, vehicles, maintenance and repair bases, power supply and transformation, communication signals, command and control centers, etc. System ordering is a key to ensure the system’s sustainable development and an important indicator to measure the internal operational quality of the system.

Some researchers have focused on safety issues about the sustainable operation of rail transit system [3,4,5,6,7,8]. For example, He et al. [3] established a quantifiable safety assessment system and equipment quality index model to strengthen quality controls for security and equipment and implemented sustainable development measures; Zhao et al. [4] are designed to investigate the factors affecting rail transit ridership at both station level and station-to-station level. Furthermore, many researchers have focused on risk factors in the sustainable operation of rail transit [5,6,7,8]. Additionally, some scholars also focus on energy conservation strategies and service quality levels [9,10,11,12,13]. Such as, Yang et al. [9] proposed a timetable optimization model due to pay more attention to the service quality; Huang [10] formulated a two-objective model to optimize the timetables of urban rail transit systems based on energy-saving strategies and service quality levels; Su et al. [11] reduced operation costs and energy consumption by optimizing the timetable and the speed profiles among successive stations; Xu et al. [12] considered the service quality and energy efficiency and developed a multi-objective timetable optimization approach for subway system. Beyond that, Zhang and Wang [16] recognized the spatial effect by estimating the ridership of the new Second Avenue Subway in New York City using a network Kriging method; Feng et al. [17] present a multilayer model by considering the characteristics of traffic flows through the network to realize the sustainable operation.

Recently, customer demand is particularly important in the study of sustainable operation of rail transit. Soltanpour et al. [14] considered the customer satisfaction and quantified passenger satisfaction of transport services; Wang et al. [15] assessed service quality and customer satisfaction on rail transit passengers’ reuse intention. They believe that sustainable operation of rail transit system must meet customer requirements to the greatest extent.

### 2.2. Related Methods

#### 2.2.1. PSR (Pressure–State–Response)

When it comes to sustainability, PSR model is a typical model for evaluation sustainable development. The PSR (pressure–state–response) model was started by Canadian statistician David J. Rapport and Anthony Marcus Friend in 1979. The PSR framework is based on the notion of causality, which is built upon the selection and measurement of indicators for three categories, i.e., indicators of pressures, of states and of responses [18]. Stress indicators illustrate the environmental impact of human and social activities and address the question “what happened?” Pressure indicators reflect environmental conditions and the changes due to human factors and address the question of “why did it happen?” Response indicators represent the remedial measures that society implements to change the environment and address the question of “what should we do?” [19]. Xie et al. [20] used the PSR model to analyze the impact of port construction on the surrounding environment; Bal-Doma´nska et al. [21] applied the PSR model to analyze socio-economic issues and spatiotemporal changes for sustainable development; Ma et al. [22] constructed a comprehensive evaluation index system for sustainable forest development and evaluated the developed level of China’s forest ecosystem. However, few scholars have applied the PSR model to the sustainable operation of rail transit system.

Based on different research objectives and requirements, the PSR model can be changed and adjusted appropriately, so as to be applied to the construction of a variety of indicator systems. PSR model can clearly show the dynamic relationship among various indices, which is conducive to the comprehensive and dynamic evaluation of systems. Based on PSR model, it is feasible to change the PSR model appropriately for constructing evaluation model and indices. Therefore, this study will establish the evaluation model and indices of sustainable operation of rail transit system based on PSR, which can clearly describe the dynamic relationship between customer requirements and operational strategies.

#### 2.2.2. QFD

As a customer-driven product design or operation strategy, the basic idea of traditional QFD method is applied to transform customers’ needs into technical attributes, engineering characteristics or operational strategies by the house of quality (HoQ) [23,24]. To date, QFD has been successfully applied in many fields, not only manufacturing [25,26], supply chain [27,28,29], healthcare service [30] but also transportation [31,32], etc.

QFD method is based on a tool called “house of quality (HoQ)”, which consists of several blocks as explained in the following [24]:The left side—customer requirements (CRs);The right side—the weights of CRs;The top—technical attributes (TAs);The body—relationship between CRs and TAs;The roof—correlation among CRs and TAs;The bottom—the importance rating of TAs.

However, due to subjective judgments made by customers and experts, the traditional models are unable to sufficiently cater for the uncertainties, vagueness, ambiguities and impreciseness [33,34]. Therefore, the fuzzy QFD was developed to get the better results [35,36,37,38]. Therefore, this study mainly applies fuzzy QFD to realize the transformation from customer requirements to operational strategies in the case of uncertainties and vagueness.

From the perspective of customer requirements, various methods are researched in order to timely get the latest voice of customers, which is a concern in all fields. Such as, Wang et al. [39] transformed customer requirements into product configuration design in product design, so that the designed products can meet customer needs as much as possible; Kuo et al. [40] took black bean as an example, and turn the requirements of customers into food development technology to meet the needs of customers in the process of food development. In addition, to analyze customer requirements, researchers have come up with different methods. Such as, Kwong and Bai [41] proposed the combination method of AHP and QFD to calculate the weight of customer requirements and finally convert it into product attributes to meet customer requirements; Wang and Tseng [42] used the Bayesian method to transform customer requirements into products variant; Wang et al. [39] developed the gray rough model to analyze customer requirements; Chandha et al. [43] combined Kano model with QFD to sort out customer demand; Nahm [44] proposed PIR method and CPR method based on QFD for considering customer requirements; Hong and Feng [45] used the fuzzy dynamic clustering method to analyze and classify the customer requirements based on QFD, etc. These methods all have their own merits, therefore which one of them can be selected according to the actual application.

Moreover, due to the different degrees of customer requirements, the weight of customer requirements needs to be considered. There are lots of methods to determine the weight of customer requirements in QFD. Such as, Zhang et al. [46] used the fuzzy comprehensive evaluation method to gain the weight of customer requirements; Kwong and Bai [41] suggested a QFD method to determine the weight of customer needs is obtained by fuzzy analytical hierarchy process (AHP); Buyukozkan et al. [47] proposed the application to get the importance of customer requirements based on fuzzy analytical network process (ANP). In addition, Shiva et al. [48] combined the evidential reasoning with fuzzy QFD. However, AHP is the most common method to determine weights, it still has many problems. Such as, the workload is too large to achieve when it has too many indices; AHP method has some subjectivity, etc. Therefore, this study will use evidential reasoning to obtain the weights of customer requirements. The advantage of evidential reasoning is: It can determine the weights of customer needs in a more objective way.

#### 2.2.3. Fuzzy Clustering

Clustering analysis [45,49] is a method to cluster indicators by establishing similarity relationship based on the characteristics, degree of intimacy and similarity. Generally speaking, clustering algorithm can be divided into hierarchical clustering, partition clustering and density clustering. To date, the mainstream clustering methods include: K-means clustering algorithm [50], Gaussian mixture algorithm [51] and fuzzy clustering [52,53].

As an important technology of data mining, fuzzy clustering has gradually become an interdisciplinary and cross-field data analysis method, which has been widely used in many fields. Such as gene expression [54], automobile insurance [55], medical science [56], text mining [57]. Biju and Mythili [54] tested the applicability of fuzzy clustering image segmentation method in cDNA chip noise image segmentation; Subudhi and Panigrahi [55] presented a novel hybrid approach for detecting frauds in automobile insurance based on fuzzy clustering and various supervised classifier models; Based on fuzzy clustering, Lewis et al. [56] proposed a method to identify abnormal neurological events associated with acute brain injuries and seizures. However, few scholars have applied fuzzy clustering to the sustainable operation of rail transit system. As analyzing customer requirements in sustainable operation of rail transit, we discover that customer requirements have different characteristics. Using the fuzzy clustering method to classify customer requirements according to their characteristics, therefore appears to be a productive method.

#### 2.2.4. Evidential Reasoning

Evidential reasoning (ER) is the method to deal with uncertainty based on the theory of D–S, which was proposed by Yang et al. [58], and was improved by Yang and Xu of the university of Manchester [59], etc. By integrating the attributes of multi-level indices and preserving the uncertainty of the initial information, the fusion decision analysis is realized to solve the mixed multi-indicator and uncertainty decision problems. This theory is widely applied to decision making [60,61], network analysis [62], optimization [63] and reliability and risk analysis [64]. After that, in order to solve the fuzziness of evaluation grade, the evaluation grade of the traditional ER method was extended to the fuzzy evaluation grade [59]. Owing to its outstanding performance in uncertainty model and process, this study uses the method of evidential reasoning to gain the weights of customer requirements and because evidential reasoning has the ability to deal with ignorance and lack of information, and more specifically, this method can provide explicit estimates of inaccuracies and inconsistencies in information in different data sets.

## 3. Preliminaries

In this section, some necessary concepts and basic knowledge are introduced which related to fuzzy set theory (Kong [57]) and Dempster–Shafer evidence theory (Fu and Yang [61]; Yang et al. [58]).

### 3.1. Fuzzy Set Theory

**Definition** **1***(Fuzzy set [57]) Let*U*denote a universal set. Then a fuzzy subset*A*of*U*is defined by its membership function:*μ(x): U→[0,1]*which assigns to each element*x∈U*a real number in the interval [0,1], where the value of*μ*(*x*) at x represents the grade of membership of*x*in*A*. Thus, the nearer the value of*μ*(*x*) is unity, the higher the grade of membership of*x*in*A.

**Definition** **2**
*(α-level of fuzzy set [57]) Let*
B
*be a fuzzy set with membership function*
B(x)
*. Then the set:*
$$B_{\alpha }=\{x\in R|B\left(x\right)\geq \alpha \}$$
*is called the*α*-level of*B*. In particular, the set*{x∈R|B(x)>0}*is called the support of*B.

**Definition** **3***(Fuzzy number [57]) Fuzzy numbers are special cases of fuzzy sets, described by given intervals of crisp numbers. Let*N*is a trapezoidal fuzzy number, denoted by*N=(a,b,c,d), whose membership functions is defined as:$$N\left(x\right)=\{\begin{array}{c} \frac{x-a}{b-a},\;if\;a\leq x\leq b,\\ 1,\;if\;b\leq x\leq c,\\ \frac{c-x}{c-d},\;if\;c\leq x\leq d,\\ 0,\;if\;else. \end{array}$$*Interval fuzzy numbers and triangular fuzzy numbers can be regarded as special cases of trapezoidal fuzzy numbers. Such as,*N*is a triangular fuzzy number when*b=c, N*is an interval fuzzy number when*a=b*and*c=d*. In particular,*N*is a crisp number when*a=b=c=d*, crisp numbers can also be considered as a special case of fuzzy numbers.*

### 3.2. Dempster–Shafer Evidence Theory

Let U denote a universal set. Then a fuzzy subset A of U is defined by its membership function:

**Definition** **4***(Basic concepts of D–S theory [58]) Suppose*$\Theta =\left\{\theta_{1},\;\theta_{2},\ldots ,\;\theta_{N}\right\}$*is a set of mutually exclusive and collectively exhaustive propositions, with*$\theta_{i}\cap \theta_{j}=\emptyset$*for any*i,j∈{1,2,…,N}*and*i≠j*where*∅*is an empty set.*Θ*is then refered to as a frame of discernment. A Basic Probability Assignment (bpa) is a function*$M:2^{\Theta }\to \left[0,1\right]$*, satisfying*M(∅)=0*and*$\underset{\theta \subseteq \Theta }{\sum }M(\theta )\;=1$.

**Definition** **5**
*(Dempster’s rule [58]) With two pieces of independent and fully reliable evidence represented by two bpas*
$M_{1}$
*and*
$M_{2}$
*respectively, for any proposition*
θ⊆Θ
*, Dempster’s rule is given as follows:*
$$M\left(\theta \right)=\left[M_{1}⊕M_{2}\right]\left(\theta \right)=\{\begin{array}{c} 0,\;\theta =\Theta ,\\ \frac{\sum_{B\cap C=\theta }M_{1}\left(B\right)M_{2}\left(C\right)}{1-\sum_{B\cap C=\Theta }M_{1}\left(B\right)M_{2}\left(C\right)},\;\theta \neq \Theta . \end{array}$$

*It is obvious in the above equation that Dempster’s rule provides a process for combining two pieces of noncompensatory evidence, in the sense that if either of them completely opposes a proposition, the proposition will not be supported at all, no matter how strongly it may be supported by the other piece of evidence.*


## 4. Evaluating the Sustainable Operation of Rail Transit System

With the formation of urban rail transit network, the traffic keeps increasing. For rail transit system, it has become the operational philosophy of rail transit system to provide safe, fast, comfortable and high-quality services for passengers and to promote the sustainable development of urban rail transit through standardized operation. In this section, the sustainable operation of rail transit system is taken as the evaluation object to evaluate the sustainable operation of rail transit system and effectively combine the customer demand with the operational strategy. At first, this study constructs the evaluation model and indices based on PSR; and then, the three-phase evaluation method is proposed to obtain the operational strategy, which is negative entropy flow.

### 4.1. The Evaluation Model and Indexes of the Sustainable Operation of Rail Transit System Based on PSR

In the rail transit system, due to the problems in service level, operation capacity and social environment, customer requirements cannot be satisfied, and positive entropy flow will be generated. The increasing entropy brings chaos to the existing system, which will weaken the sustainability of the system’s operation. The negative entropy flow can offset the increasing entropy of the system, which makes it have the characteristics of dissipative structure. These characteristics of dissipative structure can maintain the sustainable operation of rail transit system. Therefore, the negative entropy flow needs to be obtained. This study will describe the operation of rail transit system by the model of PSR and find the negative entropy flow, so as to achieve sustainable operation of rail transit system.

The PSR model is divided into three kinds of indices, namely pressure indices, state indices and response indices. In the rail transit system, pressure indices are viewed as customer requirements, which answer the reasons for such changes in this system; state indices refer to the state and environment of system activities, which can be described as the challenges of coping with pressure; “response” describes the system’s actions to address the challenges posed by customer needs, namely operational strategies. In this model, the pressure of customer requirements is the main cause of positive flow. If reasonable and effective strategies are adopted to act on this system, customer requirements will be satisfied. At that moment, the negative entropy flow will be obtained, and system chaos is reduced. Therefore, the reasonable and effective strategies in consideration of customer needs should be proposed to offset the increasing entropy, which can realize the sustainability operation of rail transit system. The PSR model is designed as shown in Figure 1 and the specific indices are set as follows:

(1). Pressure indices

Due to the limited energy of environmental resources and managers in the operation process, it is customer requirements that become the pressure on the sustainable operation of the system. In order to obtain the information of customer requirements, this study collects them by means of operational data, the sources of network, the hotline of service supervisor, interview, etc. In the process of gathering customer requirements, we find that passengers on the subway can be subdivided into different customer groups by ages, regions, population density, travel reasons, travel time, etc. By ages, it can be divided into: child, youth, adult, old age, etc.; By region, it can be divided into: local, non-province, foreign country, non-local, etc.; By population density, it can be divided into: downtown, suburbs, exurbs, etc.; By travel reasons, it can be divided into: work, school, business, personal shopping, sightseeing, visiting relations, etc.; By travel time, it can be divided into: weekday day, weekend day, weekday night, weekend night, etc. However, not all customer groups are attractive to rail transit system, for the reason that the main customer requirement groups are determined by Mckinsey Matrix, which are: child, youth, adult, old age, local, downtown, work, school, business, personal shopping, sightseeing, weekday day, weekend day, weekday night and weekend night. Through these different groups, pressure indices are summarized, which are shown in Table 1:

(2). State indices

In the investigation of customer requirements, we find that the different groups have different emphasis, such as: the customer groups of “child” and “old age” have emphasis on security needs and focus on “stable operation”, “security of closing and opening doors”, etc.; the customer groups of “youth” and “work” pay attention to the convenience and timeliness of travel, etc. Based on that, facing to the pressure of customer requirements, five state indices reflect the challenges of the sustainable operation of rail transit system, which are shown in Table 2.

(3). Response indices

Response indices, that is, operational strategies. To copy with the pressure of customer requirements and the challenges faced by the sustainable operation of rail transit system, it need to strengthen the management of technology, operations, services, resources and cost. For technical management, this system can consider from design standards and informatization; For operational management, the system can discuss from operational rules, passenger flow control, emergency linkage, etc.; For service management, this system can analyze from service process and training standards; For resource and cost management, the system can consider from subway facilities, the pricing mechanism and ticket card management. Due to that, response indices can be summarized in Table 3.

### 4.2. The Three-Phase Evaluation Method of the Sustainable Operation of Rail Transit System Based on QFD

Quality function deployment (QFD) [35,36] is used to deeply analyze customer requirements (CRs) to meet the market and customers, and then to transform them into technical attributes (TAs).

By the evaluation model and indices of rail transit system in Section 4.1, this study first, takes customer requirements (pressure indices, Ps) as input variables and obtains state indices (Ss) from Ps. Moreover, then, based on QFD, this study takes state indices (Ss) as CRs and response indices (Rs) as TAs, the rating of Rs can be obtained and the reasoning and effective operational strategies are provided, that is negative entropy flow. Moreover, then, the three-phase evaluation method is proposed. The evaluation methods include fuzzy clustering analysis, evidential reasoning, fuzzy weighted average and expected value. Moreover, the detailed steps are shown in Figure 2.

In the evaluation method based on QFD, we need to collect the data to obtain the weights of Ss and to determine the relationship matrix between Ss and Rs. In the process of data collection, we conduct a questionnaire survey to get pressure indices, according to the customer groups as following: child, youth, adult, old age, local, downtown, work, school, business, personal shopping, sightseeing, weekday day, weekend day, weekday night and weekend night. Meanwhile, interviews and questionnaires were conducted with experts to analyze “the characteristics and features of Ps” and “the relationship between Ss and Rs”. Based on these data, this study will obtain Ss from Ps by fuzzy clustering analysis, determine the weights of Ss by evidential reasoning and calculate the importance of Rs by fuzzy weighted average and expected value operator. Therefore, the three-phase sustainable operational evaluation method of rail transit system is developed:

**Phase 1.** Obtaining Ss from Ps by fuzzy clustering analysis

In this phase, fuzzy clustering method can be used to classify customer requirements to reduce the complexity of calculation.

**Step 1a.** Data standardization

Suppose that $S^{*}$ state indices are used to describe $P^{*}$ pressure indices, which denoted by $P_{i}=\{P_{i1},P_{i2},\ldots ,P_{iS^{*}}\},\;i=1,2,\ldots ,P^{*}$ and each $P_{i}$ has $S^{*}$ kinds of indicators. Based on questionnaires to experts, taking $P^{*}$ pressure indices as the row vector to get the requirements of original fuzzy matrix $\bar{P}={\left(P_{ik}\right)}_{P^{*}\times S^{*}}$, $i=1,2,\ldots ,P^{*}$, $k=1,2,\ldots ,S^{*}$, where $P_{ik}$ is a measure value of the ith pressure index described by kth state index. To meet the requirement of original fuzzy matrix, we need data conversion to narrow the data into the interval [0,1], which denoted by $P^{\prime }={\left(P_{ik}^{\prime }\right)}_{P^{*}\times S^{*}}$. The conversion formula used standard deviation [45] as follows:
(1)$$P_{ik}^{\prime }=\frac{P_{ik}-{\bar{P}}_{k}}{S_{k}}$$
where,
$${\bar{P}}_{k}=\frac{1}{P^{*}}\overset{P^{*}}{\underset{i=1}{\sum }}P_{ik},\;S_{k}{}^{2}=\frac{1}{P^{*}}\overset{P^{*}}{\underset{i=1}{\sum }}\left(P_{ik}-{\bar{P}}_{k}\right),\;k=1,2,\ldots ,S^{*}.$$

After that, if $P_{ik}^{\prime }$ is not in the interval of [0,1], the other formula used range should be carried out as follows:(2)$$P_{ik\;}^{\prime \prime }=\frac{P_{ik}^{\prime }-\underset{1\leq i\leq P^{*}}{\min}\left\{P_{ik}^{\prime }\right\}}{\underset{1\leq i\leq P^{*}}{\max}\left\{P_{ik}^{\prime }\right\}-\underset{1\leq i\leq P^{*}}{\min}\left\{P_{ik}^{\prime }\right\}},\;k=1,2,\ldots ,S^{*}.$$

At this time, it is obvious that each $P_{ik}^{\prime \prime }$ is in the interval of [0,1], so the fuzzy matrix is obtained denoted by $A={\left(P_{ik}^{\prime \prime }\right)}_{P^{*}\times S^{*}}$.

**Step 1b.** Establish fuzzy similar matrix.

To research the similarity of customer requirements between $P_{i}$ and $P_{j}$, the method of correlation coefficient is used as follows [45]:(3)$$r_{ij}=\frac{\sum_{k=1}^{S^{*}}\left(P_{ik}^{\prime \prime }-\bar{P_{i}^{\prime \prime }}\right)\left(P_{jk}^{\prime \prime }-\bar{P_{j}^{\prime \prime }}\right)}{\sqrt{\sum_{k=1}^{S^{*}}{\left(P_{ik}^{\prime \prime }-\bar{P_{i}^{\prime \prime }}\right)}^{2}\cdot {\left(P_{ik}^{\prime \prime }-\bar{P_{j}^{\prime \prime }}\right)}^{2}}}$$
where,
$$\bar{P_{i}^{\prime \prime }}=\frac{1}{S^{*}}\overset{S^{*}}{\underset{k=1}{\sum }}P_{ik}^{\prime \prime },\;\bar{P_{j}^{\prime \prime }}=\frac{1}{S^{*}}\overset{S^{*}}{\underset{k=1}{\sum }}P_{jk}^{\prime \prime }.$$

Therefore, the fuzzy similar matrix is obtained, which denoted by $T={\left(r_{ij}\right)}_{P^{*}\times P^{*}}$, where $r_{\mathrm{ij}}$ is the correlation coefficient between the ith index $P_{i}$ and the jth index $P_{j}.$ If the correlation coefficient of two requirements is larger, their characteristic are closer.

**Step 1c.** Fuzzy equivalence matrix and cluster analysis

The reason seems to be obvious that the fuzzy matrix *T* has reflexivity and symmetry but not necessarily has transitivity, the fuzzy equivalent matrix should be calculated by taking the Quadratic method [45]. we start from fuzzy similar matrix T, as follows,
$$T^{2}=T\cdot T=\overset{S^{*}}{\underset{k=1}{\vee }}\left(r_{ik}\wedge r_{jk}\right)$$

$T^{2}$ is the synthesis matrix of R. In this way, $\to T^{2}\to \cdots \to T^{2n}$, and there exist a value N, which makes
$$T^{2N}=T^{2N}\cdot T^{2N}=T^{2N+1}.$$

Then the fuzzy equivalence matrix $T^{*}$ is the transitive closure $t\left(T\right)=T^{2N}$, that is $T^{*}=T^{2N}$.

In addition, we can classify the pressure indices through the cut relation of $T^{*}$. For random λ (0≤λ≤1); we get different λ*-cut* relation $T_{ij}^{\lambda }$, the formula as follows:$$T_{ij}^{\lambda }=\{\begin{array}{c} 1,\;r_{ij}\geq \lambda ,\\ 0,\;r_{ij}<\lambda . \end{array}$$

The different $T_{ij}^{\lambda }$ can be acquired when we vary the value λ. Moreover, according to that, the right classification can be gained when the proper value λ is taken.

**Phase 2.** Calculate the weights of Ss by evidential reasoning

In this phase, evidential reasoning method will be used to evaluate the sustainable operation of rail transit system. The five state indices are obtained by the classified result from Phase 1, which are safety index, reliability index, convenience index, comfort index and economy index. The specific progress is as follows [65]:

**Step 2a.** Collecting the initial information of evaluation

Suppose that each state index $S_{m}$ (m = 1,2,…,$\;S^{*}$) is composed of $L_{m}$ pressure index denoted by $e_{ml}\left(l=1,2,\ldots ,L_{m}\right)$, which is seen as a simple secondary index layer. The relative weight of $e_{ml}$ is shown as $\omega_{ml}$, and $\overset{L}{\underset{l=1}{\sum }}\omega_{ml}=1$, $\omega_{ml}\left(l=1,2,\ldots ,L_{m}\right)\geq 0$. Let pressure index $e_{ml}$ can be evaluated at five different grades described by $H=\left\{H_{1},\ldots ,H_{n},\ldots ,H_{5}\right\}$, which are very important, important, moderate, less important and unimportant. The original assessment set of $e_{ml}$ is *SS*($e_{ml}$) *=* {($H_{n},\beta_{n,ml}$)$,n=1,2,\ldots ,5,\;l=1,2,\ldots ,L_{m}$}, the belief degree $\beta_{n,ml}$ represents the likelihood that the index $e_{ml}$ is assessed to $H_{n}$.

By investigation and collection, the single evaluation of each pressure index given by customers, the belief degree $\beta_{n,ml}$ is obtained by:$$\beta_{n,ml}=\frac{q_{n,ml}}{Q_{ml}}$$
where, $q_{n,ml}$ represents the number of customers who assess the pressure index $e_{ml}$ to the degree $H_{n}$ and $Q_{ml}$ represents the total number of customers who assess $e_{ml}$ in the evaluation.

**Step 2b.** Calculate the assessment set of Ss.

By the theory of D–S, the belief degree $\beta_{n,m}$ which represents the state index $S_{m}$ (m = 1,2,…,$\;S^{*}$) is assessed to the degree $H_{n}$ is synthesized. See Formulate (4)–(12):(4)$$M_{n,ml}=\omega_{ml}\beta_{n,ml},$$
(5)$$M_{H,ml}=1-\omega_{ml}\sum_{n=1}^{5}\beta_{n,ml},$$
(6)$${\bar{M}}_{H,ml}=1-\omega_{ml},{\tilde{M}}_{H,ml}=\omega_{ml}\left(1-\sum_{n=1}^{5}\beta_{n,ml}\right),$$
(7)$$M_{H,ml}={\bar{M}}_{H,ml}+{\tilde{M}}_{H,ml}.$$
where, $M_{n,ml}$ be a basic probability mass, $M_{H,ml}$ be a remaining probability mass, ${\bar{M}}_{H,ml}$ represents the degree to which other indices can play a role in the assessment, ${\tilde{M}}_{H,ml}$ is caused due to the incompleteness in the assessment *SS*($e_{ml}$), S*S*($e_{ml}$) will be zero if S*S*($e_{ml}$) is complete.

Next, the combined probability masses *SS*($e_{ml}$) are generated by aggregating the assessments S*S*($e_{mi}$) and S*S*($e_{mj}$)as follows:(8)$$M_{n,mI\left(l+1\right)}=K_{mI\left(l+1\right)}\left(M_{n,mI\left(l\right)}M_{n,m\left(l+1\right)}+M_{H,mI\left(l\right)}M_{n,m\left(l+1\right)}+M_{n,mI\left(l\right)}M_{H,m\left(l+1\right)}\right),$$
(9)$${\bar{M}}_{H,mI\left(l+1\right)}=K_{mI\left(l+1\right)}{\bar{M}}_{H,mI\left(l\right)}{\bar{M}}_{H,m\left(l+1\right)},$$
(10)$$M_{H,mI\left(l+1\right)}={\tilde{M}}_{H,mI\left(l+1\right)}+{\bar{M}}_{H,mI\left(l+1\right)},$$
(11)$$K_{mI\left(l+1\right)}={\left(1-\sum_{t=1}^{5}\sum_{j=1,j\neq t}^{5}M_{t,mI\left(l\right)}M_{j,m\left(l+1\right)}\right)}^{-1}.$$
where, $M_{n,mI\left(l+1\right)}$ is the combined probability mass for the grade $H_{n}$ by aggregating l+1 assessments for the state index $S_{m}$. Finally, the formula of assessments *SS*($S_{m}$) as follows:(12)$$\beta_{n,m}=\frac{M_{n,mI\left(L_{m}\right)}}{1-{\bar{M}}_{H,mI\left(L_{m}\right)}}\;.$$

**Step 2c.** Calculate the weights of Ss.

Let the five different grades denoted by $H=\left\{H_{1},\ldots ,H_{n},\ldots ,\;H_{5}\right\}$, (*n* = 1,2,…,5), which belongs to a predefined triangular fuzzy set $\{H_{1}^{*},\ldots ,H_{s}^{*},\ldots ,H_{q}^{*}\},\;s=1,2,\ldots ,q$, that is, $H_{n}\in \{H_{1}^{*},\ldots ,H_{s}^{*},\ldots ,H_{q}^{*}\}$. Therefore, the weights of state indices are as follows:(13)$$W_{m}=\sum_{n=1}^{5}H_{n}\beta_{n,m}.$$

It is obvious that $W_{m}$ is a triangular fuzzy number.

**Phase 3.** Rating the importance of Rs by fuzzy weighted average and expected value

Vangeas and Labib [66] first suggested the use of fuzzy weighted average in QFD and proposed a model for deriving optimum targets of TAs through the implementation of fuzzy weighted average, whose membership functions are nonlinear in essence and is not explicitly known in most cases. This method can hardly be applied while the derived membership function of fuzzy weighted average is not explicitly known. In order to overcome the above problem, Chen [35] et al. proposed the method by integrating the fuzzy weighted average method and the fuzzy expected value operator, which is so effective that it is widely used. The specific progress is as follows:

**Step 3a.** Collecting the data

Based on the weight of each state index obtained by the last phase, the fuzzy relationship $U_{mj}$ between $S_{m}\left(m=1,2,\ldots ,S^{*}\right)$ and $R_{j}\left(j=1,2,\ldots ,R^{*}\right)$ is just determined. The calculating formulate as follows:(14)$$U_{mj}=\frac{1}{b}\sum_{v=1}^{b}U_{mj}^{v},\;m=1,2,\ldots ,S^{*},j=1,2,\ldots ,R^{*}.$$
where, $U_{mj}^{v}$ represents the fuzzy relationship between mth state index $S_{m}$ and jth operation index $R_{j}$, which belongs to a predefined triangular fuzzy set $\left\{U_{1}^{*},\ldots ,U_{t}^{*},\ldots ,U_{p}^{*}\right\}$, t=1,2,…,p, that is $U_{mj}^{v}\in \{U_{1}^{*},\ldots ,U_{t}^{*},\ldots ,U_{p}^{*}\}$; *v* is the number of experts participated in the survey.

**Step 3b.** Calculating the fuzzy importance of Rs

Let $W_{m}=\left\{w_{i},\mu_{W_{m}}\left(w_{m}\right)|w_{m}\in {\tilde{W}}_{m}\right\},m=1,2,\ldots ,S^{*},\;U_{mj}=\left\{u_{mj},\mu_{U_{mj}}\left(u_{mj}\right)|u_{mj}\in {\tilde{U}}_{mj}\right\},m=1,2,\;\ldots ,S^{*},\;j=1,2,\ldots ,R^{*}.$ Where $\mu_{W_{m}}\left(w_{m}\right)$ and $\mu_{U_{mj}}\left(u_{mj}\right)$ are the membership functions of $\;W_{m}$ and $U_{mj}$ respectively, ${\tilde{W}}_{m}$ and $U_{mj}$ are crisp number sets. Based on the fuzzy weighted average method, the fuzzy importance of Rs can be obtained as follows:(15)$$Y_{j}=\frac{\sum_{i=1}^{S^{*}}W_{m}U_{mj}}{\sum_{m=1}^{S^{*}}W_{m}},\;j=1,2,\ldots ,R^{*};$$
It is evident that $Y_{j}$ is a triangular fuzzy number.

Then, the fuzzy weighted average $Y_{j}$ is defined as the following membership function $\mu_{Y_{j}}\left(y_{j}\right)$, with respected to the fuzzy extension principle [35]:(16)$$\mu_{Y_{j}}\left(y_{j}\right)=\underset{u,w}{max}\;min\left\{\mu_{W_{m}}\left(w_{m}\right),\mu_{U_{mj}}\left(u_{mj}\right),\forall m,j|y_{j}=\frac{\sum_{m=1}^{S^{*}}w_{m}u_{mj}}{\sum_{m=1}^{S^{*}}w_{m}}\right\};$$

Clearly, the equation above can be converted into the equivalent NLP model as follows:$$\mu_{Y_{j}}\left(y_{j}\right)=\max\;z$$
(17)$$s.t.\{\begin{array}{c} \;z\leq \mu_{W_{m}}\left(w_{m}\right),\;m=1,2,\ldots ,S^{*};\\ \;z\leq \mu_{U_{mj}}\left(u_{mj}\right),\;m=1,2,\ldots ,S^{*};\\ y_{j}=\frac{\sum_{m=1}^{S^{*}}w_{m}u_{mj}}{\sum_{m=1}^{S^{*}}w_{m}},\;m=1,2,\ldots ,S^{*};\\ w_{m}\in {\tilde{W}}_{m},\;u_{mj}\in {\tilde{U}}_{mj},\;m=1,2,\ldots ,S^{*}. \end{array}$$

Due to the functions $\mu_{W_{m}}\left(w_{m}\right)$ and $\mu_{U_{mj}}\left(u_{mj}\right)$ are nondifferentiable, so it is difficult to get the optimal solution in this model. In order to solve this problem, by handling the α−*cuts* of $Y_{j}$ to replace the above model. Let ${\left(W_{m}\right)}_{\alpha }$, ${\left(U_{mj}\right)}_{\alpha }$ are the α−*cuts* of $W_{m}$, $U_{mj}$, respectively:$${\left(W_{m}\right)}_{\alpha }=\left\{w_{m}\in W_{m}|\mu_{W_{m}}\left(w_{m}\right)\geq \alpha ,\;0\leq \alpha \leq 1\right\}=\left[{\left(W_{m}\right)}_{\alpha }^{L},{\left(W_{m}\right)}_{\alpha }^{U}\right],$$
$${\left(U_{mj}\right)}_{\alpha }=\left\{u_{mj}\in U_{mj}|\mu_{U_{mj}}\left(u_{mj}\right)\geq \alpha ,\;0\leq \alpha \leq 1\right\}=\left[{\left(U_{mj}\right)}_{\alpha }^{L},{\left(U_{mj}\right)}_{\alpha }^{U}\right].$$

Similarly, let ${\left(Y_{j}\right)}_{\alpha }=\left[{\left(Y_{j}\right)}_{\alpha }^{L},{\left(Y_{j}\right)}_{\alpha }^{U}\right]$ is the α−*cuts* of $Y_{j}$. Moreover, then, ${\left(Y_{j}\right)}_{\alpha }$ can be obtained by the following NLP models:(18)$$\{\begin{array}{c} {\left(Y_{j}\right)}_{\alpha }^{L}=min\frac{\sum_{m=1}^{S^{*}}w_{m}{\left(U_{mj}\right)}_{\alpha }^{L}}{\sum_{m=1}^{S^{*}}w_{m}}\\ s.t.\;{\left(W_{m}\right)}_{\alpha }^{L}\leq w_{m}\leq {\left(W_{m}\right)}_{\alpha }^{U} \end{array}$$
and
(19)$$\{\begin{array}{c} {\left(Y_{j}\right)}_{\alpha }^{U}=max\frac{\sum_{i=1}^{S^{*}}w_{m}{\left(U_{mj}\right)}_{\alpha }^{U}}{\sum_{m=1}^{S^{*}}w_{m}}\\ s.t.\;{\left(W_{m}\right)}_{\alpha }^{L}\leq w_{m}\leq {\left(W_{m}\right)}_{\alpha }^{U} \end{array}$$

Let $t=1/\overset{S^{*}}{\underset{m=1}{\sum }}w_{m},\;v_{m}=tw_{m}$, the models of (3–5) and (3–6) can be transformed into the following LP models:$${\left(Y_{j}\right)}_{\alpha }^{L}=min\overset{S^{*}}{\underset{m=1}{\sum }}v_{m}{\left(U_{mj}\right)}_{\alpha }^{L}$$
(20)$$s.t.\;\{\begin{array}{c} t{\left(W_{m}\right)}_{\alpha }^{L}\leq tv_{m}\leq t{\left(W_{m}\right)}_{\alpha }^{U},\\ \sum_{m=1}^{S^{*}}v_{m}=1,\\ t,\;v_{m}\geq 0;m=1,2,\ldots ,S^{*};j=1,2,\ldots ,R^{*}. \end{array}$$
and
$${\left(Y_{j}\right)}_{\alpha }^{U}=max\overset{S^{*}}{\underset{m=1}{\sum }}v_{m}{\left(U_{mj}\right)}_{\alpha }^{U}$$
(21)$$s.t.\;\{\begin{array}{c} t{\left(W_{m}\right)}_{\alpha }^{L}\leq tv_{m}\leq t{\left(W_{m}\right)}_{\alpha }^{U},\\ \sum_{m=1}^{S^{*}}v_{m}=1,\\ t,\;v_{m}\geq 0;m=1,2,\ldots ,S^{*};j=1,2,\ldots ,R^{*}. \end{array}$$

Taking the different value α, the following approximate membership function $\mu_{Y_{j}}\left(y_{j}\right)$ of operation index $Y_{j}$ can be calculated by the models of (20) and (21):(22)$$\mu_{Y_{j}}\left(y_{j}\right)=\{\begin{array}{c} L\left(y_{j}\right),\;{\left(Y_{j}\right)}_{\alpha =0}^{L}\leq y_{j}\leq {\left(Y_{j}\right)}_{\alpha =1}^{L}\\ 1,\;{\left(Y_{j}\right)}_{\alpha =1}^{L}\leq y_{j}\leq {\left(Y_{j}\right)}_{\alpha =1}^{U}\\ R\left(y_{j}\right),\;{\left(Y_{j}\right)}_{\alpha =0}^{U}\leq y_{j}\leq {\left(Y_{j}\right)}_{\alpha =1}^{U} \end{array}$$

**Step 3c.** Rating the importance of Rs

Based on the expected value operator, the rating of Rs can be gained by calculate the expected value $E\left(Y_{j}\right)$:(23)$$E\left(Y_{j}\right)=\frac{1}{2L}\sum_{f=1}^{L}\left({\left(Y_{j}\right)}_{\alpha_{f}}^{L}+{\left(Y_{j}\right)}_{\alpha_{f}}^{U}\right)$$
where, $\left\{\left(\alpha_{1},\;\cdots ,\alpha_{f},\cdots ,\alpha_{L}\right)|0=\alpha_{1}<\cdots <\alpha_{f}<\cdots <\alpha_{L}=1\right\}$ represents the set of α.

## 5. Empirical Analysis

This section will take customers as the main object to empirically analyze and evaluate the sustainable operation of Shanghai rail transit system based on the proposed model and methods in this study and take Shentong Metro as the object of investigation. In this research, questionnaires were used to collect and obtain relevant information. More specifically, for the data of customer requirements, 18,091 useful questionnaires were collected from the different customer groups which are child, youth, adult, old age, local, downtown, work, school, business, personal shopping, sightseeing, weekday day, weekend day, weekday night and weekend night; for the data of “the characteristics and features of basic customer requirements” and “the relationship between Ss and Rs”, 10 experts, who are professional researchers and operations managers with professional experience in rail transit operation management, participated in the questionnaire. Based on the three-phase method in Section 4.2, the process is as follows: (where $P^{*}=13,$
$S^{*}=5$, $R^{*}=11$).

**Phase 1.** Obtaining Ss from Ps by fuzzy clustering analysis

At first, the original fuzzy matrix will be collected. According to the questionnaires of “the characteristics and features of 13 pressure indices” from 10 experts (let the scores are in the interval [0,1]), the original fuzzy matrix $\bar{P}$ is obtained as follows:
$$\bar{X}=[\begin{array}{c} 0.30\;0.60\;0.70\;0.98\;0.20\\ 0.20\;0.75\;0.93\;0.60\;0.20\\ 0.95\;0.75\;0.63\;0.80\;0.20\\ 0.91\;0.60\;0.70\;0.98\;0.20\\ 0.40\;0.90\;0.75\;0.65\;0.30\\ 0.92\;0.68\;0.66\;0.76\;0.20\\ 0.20\;0.52\;0.85\;0.57\;0.90\\ 0.35\;0.88\;0.75\;0.68\;0.35\\ 0.55\;0.66\;0.70\;0.96\;0.22\\ 0.20\;0.76\;0.95\;0.65\;0.25\\ 0.20\;0.49\;0.82\;0.58\;0.95\\ 0.93\;0.72\;0.69\;0.72\;0.40\\ 0.20\;0.58\;0.83\;0.56\;0.90 \end{array}]$$


In addition, the fuzzy matrix *A*, the fuzzy similar matrix *T* and the fuzzy equivalence matrix $T^{*}$ are calculated respectively according to the fuzzy clustering analysis from Phase 1 of Section 4.2. In this case, the fuzzy equivalence matrix $T^{*}$ is just shown as follows:
$$\left[\begin{array}{c} 1.0000\;0.8531\;0.8531\;0.8531\;0.7575\;0.8531\;0.8531\;0.7575\;0.9570\;0.8531\;0.8531\;0.8531\;0.8531\\ 0.8531\;1.0000\;0.9458\;0.9458\;0.7575\;0.9458\;0.9210\;0.7575\;0.8531\;0.9944\;0.9210\;0.9458\;0.9210\\ 0.8531\;0.9458\;1.0000\;0.9867\;0.7575\;0.9867\;0.9210\;0.7575\;0.8531\;0.9458\;0.9210\;0.9867\;0.9210\\ 0.8531\;0.9458\;0.9867\;1.0000\;0.7575\;0.9867\;0.9210\;0.7575\;0.8531\;0.9458\;0.9210\;0.9893\;0.9210\\ 0.7575\;0.7575\;0.7575\;0.7575\;1.0000\;0.7575\;0.7575\;0.9863\;0.7575\;0.7575\;0.7575\;0.7575\;0.7575\\ 0.8531\;0.9458\;0.9867\;0.9867\;0.7575\;1.0000\;0.9210\;0.7575\;0.8531\;0.9458\;0.9210\;0.9867\;0.9210\\ 0.8531\;0.9210\;0.9210\;0.9210\;0.7575\;0.9210\;1.0000\;0.7575\;0.8531\;0.9210\;0.9892\;0.9210\;0.9841\\ 0.7575\;0.7575\;0.7575\;0.7575\;0.9863\;0.7575\;0.7575\;1.0000\;0.7575\;0.7575\;0.7575\;0.7575\;0.7575\\ 0.9570\;0.8531\;0.8531\;0.8531\;0.7575\;0.8531\;0.8531\;0.7575\;1.0000\;0.8531\;0.8531\;0.8531\;0.8531\\ 0.8531\;0.9944\;0.9458\;0.9458\;0.7575\;0.9458\;0.9210\;0.7575\;0.8531\;1.0000\;0.9210\;0.9458\;0.9210\\ 0.8531\;0.9210\;0.9210\;0.9210\;0.7575\;0.9210\;0.9892\;0.7575\;0.8531\;0.9210\;1.0000\;0.9210\;0.9841\\ 0.8531\;0.9458\;0.9867\;0.9893\;0.7575\;0.9867\;0.9210\;0.7575\;0.8531\;0.9458\;0.9210\;1.0000\;0.9210\\ 0.8531\;0.9210\;0.9210\;0.9210\;0.7575\;0.9210\;0.9841\;0.7575\;0.8531\;0.9210\;0.9841\;0.9210\;1.0000 \end{array}\right]$$


At the end, there is clustering. Taking the different value λ (0≤λ≤1), the classified results are:
1)It can be divided into 13 categories when λ=1: {$P_{1}$}, {$P_{2}$}, {$P_{3}$}, {$P_{4}$}, {$P_{5}$}, {$P_{6}$}, {$P_{7}$}, {$P_{8}$}, {$P_{9}$}, {$P_{10}$}, {$P_{11}$}, {$P_{12}$}, {$P_{13}$};2)It can be divided into 12 categories when 0.99<λ≤1: {$P_{1}$}, {$P_{2},P_{10}$}, $\{P_{3}$}, {$P_{4}$}, {$P_{5}$}, {$P_{6}$}, {$P_{7}$}, {$P_{8}$}, {$P_{9}$}, {$P_{11}$}, {$P_{12}$}, {$P_{13}$};3)It can be divided into 6 categories when 0.98<λ≤0.99: {$P_{1}$}, {$P_{2},P_{10}$}, $\left\{P_{3},P_{4},P_{6},P_{12}\right\},\;\left\{P_{5},P_{8}\right\}$, {$P_{9}$}, $\left\{P_{7},P_{11},P_{13}\right\}$;4)It can be divided into 5 categories when 0.95<λ≤0.98: $\left\{P_{1},P_{9}\right\},\;\left\{P_{2},P_{10}\right\},\left\{P_{3},P_{4},P_{6},P_{12}\right\},\left\{P_{5},P_{8}\right\},\;\left\{P_{7},P_{11},P_{13}\right\}$;5)It can be divided into 4 categories when 0.94<λ≤0.95: $\left\{P_{1},P_{9}\right\},\;${$P_{2},P_{3},P_{4},P_{6},P_{10},P_{12}\},\left\{P_{5},P_{8}\right\},\;\left\{P_{7},P_{11},P_{13}\right\}$;6)It can be divided into 3 categories when 0.92<λ≤0.9: $\left\{P_{1},P_{9}\right\}$, $\left\{P_{5},P_{8}\right\},\;${$P_{2},P_{3},P_{4},\;P_{6},P_{7},P_{10},$$P_{11},P_{12},P_{13}\}$;7)It can be divided into 2 categories when 0.85<λ≤0.92:$\;\{P_{1},P_{2},P_{3},P_{4},P_{6},P_{7},P_{9},P_{10},P_{11},P_{12},P_{13}\}$, $\left\{P_{5},P_{8}\right\}$;8)It can be classified 1.

When 13 pressure indices are described by 5 state indices, it is obvious that the right classification can be obtained when 0.95<λ≤0.98. Therefore, the right classification is as follows: $S_{1}=\left\{P_{3},P_{4},P_{6},P_{12}\right\}$, $S_{2}=\left\{P_{5},P_{8}\right\}$, $S_{3}=\left\{P_{2},P_{10}\right\}$, $S_{4}=\left\{P_{1},P_{9}\right\}$, $S_{5}=\left\{P_{7},P_{11},P_{13}\right\}$.

**Phase 2.** Calculate the weights of Ss by evidential reasoning.

Based on the classification of the last phase, we can get that $L_{1}=4$, $L_{2}=2$, $L_{3}=2$, $L_{4}=2$, $L_{5}=3$. Therefore, let $S_{1}=\left\{P_{3},P_{4},P_{6},P_{12}\right\}=\left\{e_{11},e_{12},e_{13},e_{14}\right\}$, $S_{2}=\left\{P_{5},P_{8}\right\}=\left\{e_{21},e_{22}\right\}$, $S_{3}=\left\{P_{2},P_{10}\right\}=\left\{e_{31},e_{32}\right\}$, $S_{4}=\left\{P_{1},P_{9}\right\}=\left\{e_{41},e_{42}\right\}$, $S_{5}=\left\{P_{7},P_{11},P_{13}\right\}=\left\{e_{51},e_{52},e_{53}\right\}$. In addition, let the relative importance of each pressure index is equal, that is $\omega_{11}=\omega_{12}=\omega_{13}=\omega_{14}=$$\frac{1}{4},{\;\omega }_{21}=\omega_{22}=\omega_{31}=\omega_{32}=\omega_{41}=\omega_{42}=$1/2, $\omega_{51}=\omega_{52}=\omega_{53}=$1/3.

At first, determining the original assessment set *SS*($e_{ml}$). Let pressure index $e_{\mathrm{mL}}$ can be evaluated at five different grades described by $H=\left\{H_{1},\ldots ,H_{n},\ldots ,H_{5}\right\}$, which are very important, important, moderate, less important and unimportant. Moreover, these are triangular fuzzy numbers which are shown respectively: $H_{1}=\left(0.75,1,1\right)$, $H_{2}=\left(0.5,0.75,1\right)$, $H_{3}=\left(0.25,0.5,0.75\right)$, $H_{4}=\left(0,0.25,0.5\right)$, $H_{5}=\left(0,0,0.25\right)$. According to the questionnaires, the original assessment set *SS*($e_{ml}$) is analyzed as shown in Table 4.

Further, based on the Equations (4)–(12) in Section 4.2, the combined assessment sets *SS*($S_{m}$) are calculated shown as Table 5:

Afterwards, the weights of 5 state indices are calculated according to the Equation (13) shown in Section 4.2: $W_{1}=\left(0.4707,\;0.7040,\;0.8483\right),\;W_{2}=\left(0.4044,\;0.6283,\;0.7675\right),\;W_{3}=\left(0.4329,\;0.6393,\;0.7776\right),\;W_{4}=\left(0.4633,\;0.7103,\;0.8647\right),\;W_{5}=\left(0.4684,\;0.6980,\;0.8600\right)$.

**Phase 3.** Rating the importance of Rs by fuzzy weighted average and expected value.

Based on Phase 1 and Phase 2, the weights of 5 state indices are obtained by analyzing 13 pressure indices. Then, this study transforms 5 state indices into 11 response indices by fuzzy weighted average and expected value.

To begin with, determining the relationship of Ss and Rs. Let the relationship of Ss and Rs are expressed as the linguistic data at seven levels, which are none, weak, moderate, strong and very strong. The corresponding triangular fuzzy set denoted as $U^{*}=\{U_{1}^{*},U_{2}^{*},U_{3}^{*},U_{4}^{*},U_{5}^{*}\}$, where $U_{1}^{*}=\left(0,0,0.3\right)$, $U_{2}^{*}=\left(0,0.25,0.5\right)$, $U_{3}^{*}=\left(0.3,0.5,0.7\right)$, $U_{4}^{*}=\left(0.5,0.75,1\right)$, $U_{5}^{*}=\left(0.7,1,1\right)$. According to the questionnaires from 10 experts, the relationship $U_{\mathrm{ij}}$ between Ss and Rs are collected and calculated.

Accordingly, the importance ranking of Rs by fuzzy weighted average and expected value operator as shown in Table 6.

From Table 3, we can get that the rankings of top 5 are: “advancement of design standards $\left(R_{1}\right)$”, “reliability of subway facilities $\left(R_{9}\right)$”, “completeness of operational rules $\left(R_{3}\right)$”, “standardization of management operation $\left(R_{5}\right)$” and “rationality of passenger flow control $\left(R_{4}\right)$”. Therefore, Shanghai rail transit system start from these five response indices and rectify the operation strategy, which will play an important role in promoting the sustainable development of rail transit operational system.

## 6. Conclusions

In rail transit system, due to the problems in service level, operation capacity, operation cost and social environment, positive entropy flow will be generated, the increasing entropy of the system will weaken the sustainable operation. Based on PSR and QFD, with the sustainable operation of rail transit system as the evaluation object, the innovation of this study is to analyze customer requirements and provide effective and reasonable operational strategies of rail transit system, namely negative entropy flow.

First of all, the evaluation model and indices established in this study are based on PSR, including 13 Ps, 5 Ss and 11 Rs. Then, based on QFD, this study proposed the three-phase evaluation method of the sustainable operation of rail transit system with 13 Ps as input variables, 5 Ss as CRs of QFD and 11 Rs as TAs of QFD. Here, the three-phase evaluation method of the sustainable operation of rail transit system is as follows: The first phase is to verify that 13 Ps can be clustered into 5 Ss by fuzzy clustering analysis; The second phase is to get the weights of 5 Ss by evidential reasoning; The third phase is to rate the importance of 11 Rs by integrating fuzzy weighted average and expected value operator. Finally, Shanghai rail transit system is taken as an example, we come to the conclusion that the raking of 11 Rs is as follows: $R_{1}>R_{9}>R_{3}>R_{5}>R_{4}>R_{2}>R_{7}>R_{8}>R_{6}>R_{11}>R_{10}$. Therefore, from the empirical analysis, the top five are: “advancement of design standards $\left(R_{1}\right)$”, “reliability of subway facilities $\left(R_{9}\right)$”, “completeness of operational rules $\left(R_{3}\right)$”, “standardization of management operation $\left(R_{5}\right)$” and “rationality of passenger flow control $\left(R_{4}\right)$”. The Shanghai rail transit system should take priority from these five aspects in the absence of resources and rectify the operation strategy. Based on these five perspectives, the following suggestions of sustainable operation can be provided:From the perspective of “advancement of design standards”, relevant departments can realize the full-automatic driving function of some routes to improve driving stability;From the perspective of “reliability of subway facilities”, relevant departments can consider the following aspects of rectification: (a) support for both online and on-site ticketing functions; (b) top of TVM displays the dynamic fare information of the whole network, the time of the first and last bus, the real-time operation status, subway announcements, emergency information, etc.; (c) equip with shielding doors, safety gates, safety barriers and infrared, laser and other detection equipment; (d) install help buttons to respond to customer requests.From the perspective of “completeness of operational rules”, relevant departments can consider the following aspects of rectification: (a) formulate standard operating rules for electric train drivers to ensure safety; (b) formulate cleaning management regulations and relevant work instructions to improve quality standards and clarify cleaning methods, process tool requirements and intensity distribution.From the perspective of “standardization of management operation”, relevant departments can use real-time data of the passenger flow to track the source, predict the subsequent trend of returning passenger in combination with sorting algorithm and to adjust the interval timely;From the perspective of “rationality of passenger flow control”, relevant departments can increase station staff and volunteers, etc.

This study considers the customer’s needs and integrates the whole system. In the future, the researchers can consider the difference and impact on the weights of customer requirements in case of different customer groups, as well as the effect on the rankings of response indices of rail transit system.

## Figures and Tables

**Figure 1 entropy-22-00750-f001:**
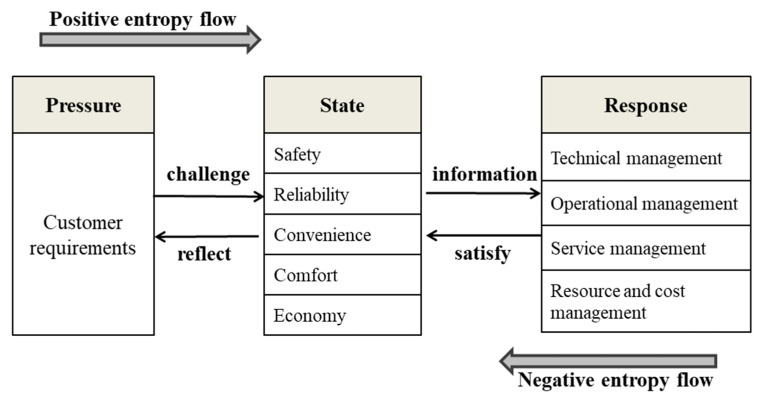
**Pressure-State-Response** (PSR) model of the sustainable operation of rail transit system.

**Figure 2 entropy-22-00750-f002:**
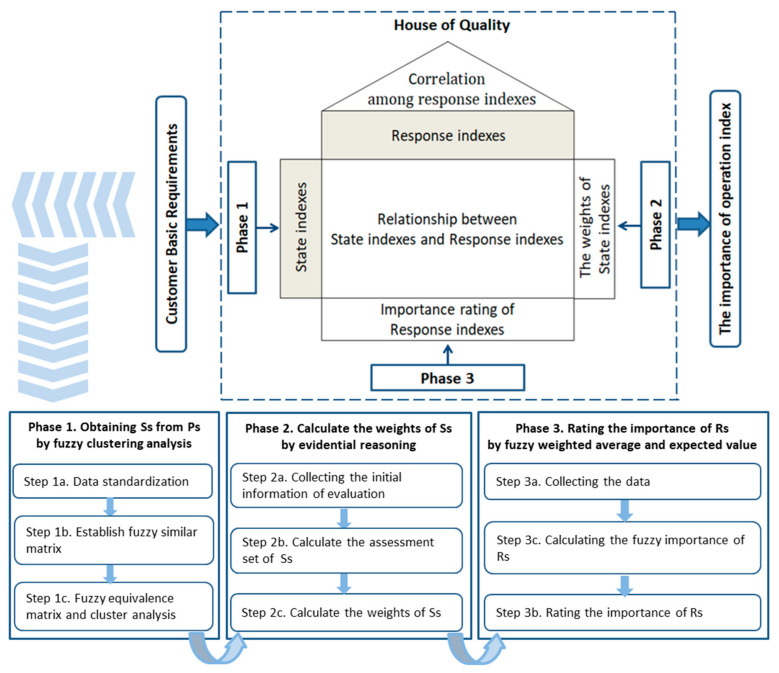
Three-phase evaluation method based on quality function deployment (QFD).

**Table 1 entropy-22-00750-t001:** Pressure indices and their explanations.

Pressure Indices	Explanation
Clearness of environment $\left(P_{1}\right)$	Including platform cleanliness, station hall cleanliness and train cleanliness.
Timely informations $\left(P_{2}\right)$	Including subway stops, delay information, operational time, stoppage time and other timely feedback.
Stable operation $\left(P_{3}\right)$	Including the train time to meet customer requirements, interior security and the safe and smooth running of the vehicle.
Good waiting orders $\left(P_{4}\right)$	Including orderly lines of passengers, getting off first and then on when the train arrives.
Less delay $\left(P_{5}\right)$	Including trains arriving on time, the absence of accidents.
No four promiscuity $\left(P_{6}\right)$	Including no booth, no litter, duty officer and security inspectors in place.
Convenient ticket purchase $\left(P_{7}\right)$	Including not taking too long to buy tickets, the number of ticket machines to meet customer requirements and multiple ways to purchase tickets.
Flexible departure interval $\left(P_{8}\right)$	Including the departure interval to meet the commuter flow and the number of departure to meet the daily needs of passenger flow.
Good services $\left(P_{9}\right)$	Including good platform facilities, good train operation, good service attitude and language standard of staff.
Easy access $\left(P_{10}\right)$	Including the number of platforms to access information, the operation of the line can be known anytime and anywhere.
Cheap fares $\left(P_{11}\right)$	Including credit card discount, long distance travel discount.
Security of closing and opening doors $\left(P_{12}\right)$	Including the safety of passengers when getting on and off the bus, the door is not clamped.
Various tickets $\left(P_{13}\right)$	Including the various ways of buying tickets, various forms of tickets, such as day tickets, multi-day tickets and so on.

**Table 2 entropy-22-00750-t002:** State indices and their explanations.

State Indices	Explanation
Safety $\left(S_{1}\right)$	Ensure that passengers will not cause psychological and physiological damage.
Reliability $\left(S_{2}\right)$	Ensure that passengers arrive at their destinations.
Convenience $\left(S_{3}\right)$	Ensure the efficiency of passenger transfer and accurately identify entrances and exits.
Comfort $\left(S_{4}\right)$	Create a clean and comfortable environment for passengers.
Economy $\left(S_{5}\right)$	Ensure the reasonableness of ticket prices.

**Table 3 entropy-22-00750-t003:** Response indices and their explanations.

	Response Indices	Explanation
Technical management	advancement of design standards $\left(R_{1}\right)$	Design standards refer to the subway design, operational definition of related principles and regulations, such as metro design principles, methods, parameters, limit, equipment installation and acceptance, operation organization, operating personnel, vehicles, emergency drills, alignment, etc.
	Popularization of informatization $\left(R_{2}\right)$	Informatization refers to the use of modern communications, network, database technology, in order to improve service efficiency.
Operational management	Completeness of operational rules $\left(R_{3}\right)$	Operational rules refer to the establishment of rules and regulations for the organization operating process and operation management.
	rationality of passenger flow control $\left(R_{4}\right)$	Passenger flow control refers to the effective control of the station passenger flow limit.
	standardization of management operation $\left(R_{5}\right)$	Management operation refers to the daily management operation standards of the station post work, work standards and procedures as well as the station traffic, passenger transport, station, etc.
	Coordination of emergency linkage $\left(R_{6}\right)$	Emergency linkage refers to a series of emergency measures taken in the process of subway operation.
Service management	Rationality of service process $\left(R_{7}\right)$	Service process refers to the specific requirements and standards of customer service.
	Suitability of training standards $\left(R_{8}\right)$	Training standards set up the subway service personnel training standards.
Resource and cost management	reliability of subway facilities $\left(R_{9}\right)$	Subway facilities refer to the equipment used in the subway station. The subway facility equipment system is divided into the basic facility system and the operating equipment system: basic facility system includes lines, tracks, stations, etc. operating equipment system include vehicles, vending machines, air conditioners, escalators, etc.
	Innovation of the pricing mechanism $\left(R_{10}\right)$	Pricing mechanism refers to the setting of principles, methods, charging standards, calculation models and preferential policies for subway fares.
	Order of ticket card management $\left(R_{11}\right)$	Ticket card management includes the purchase and management of all kinds of ticket cards and the flow of each link in the online network.

**Table 4 entropy-22-00750-t004:** Original assessment sets SS($e_{\mathrm{mL}}$) of 13 pressure indices.

State Indices	Pressure Indices	Original assessment sets SS $\left(e_{ml}\right)$
Safety $\left(S_{1}\right)$	Stable operation $\left(e_{11}\right)$	$\left\{\begin{array}{c} \left(H_{1},0.5268\right),\;\left(H_{2},0.2470\right),\;\left(H_{3},0.1957\right),\\ \;\left(H_{4},0.0143\right),\left(H_{5},0.0161\right) \end{array}\right\}$
	Good waiting order $\left(e_{12}\right)$	$\left\{\begin{array}{c} \left(H_{1},0.4422\right),\;\left(H_{2},0.2543\right),\;\left(H_{3},0.2149\right),\;\\ \left(H_{4},0.0333\right),\left(H_{5},0.0553\right) \end{array}\right\}$
	No four promiscuity $\left(e_{13}\right)$	$\left\{\begin{array}{c} \left(H_{1},0.4974\right),\;\left(H_{2},0.3616\right),\;\left(H_{3},0.0675\right),\;\\ \left(H_{4},0.0165\right),\left(H_{5},0.0569\right) \end{array}\right\}$
	Security of closing and opening doors $\left(e_{14}\right)$	$\left\{\begin{array}{c} \left(H_{1},0.3008\right),\;\left(H_{2},0.1891\right),\;\left(H_{3},0.1665\right),\;\\ \left(H_{4},0.2220\right),\left(H_{5},0.1217\right) \end{array}\right\}$
Reliability $\left(S_{2}\right)$	Clearness of environment $\left(e_{21}\right)$	$\left\{\begin{array}{c} \left(H_{1},0.3807\right),\;\left(H_{2},0.1782\right),\;\left(H_{3},0.0111\right),\;\\ \left(H_{4},0.3726\right),\left(H_{5},0.0573\right) \end{array}\right\}$
	Good service $\left(e_{22}\right)$	$\left\{\begin{array}{c} \left(H_{1},0.4594\right),\;\left(H_{2},0.1112\right),\;\left(H_{3},0.0294\right),\;\\ \left(H_{4},0.2278\right),\left(H_{5},0.1722\right) \end{array}\right\}$
Convenience $\left(S_{3}\right)$	Less delay $\left(e_{31}\right)$	$\left\{\begin{array}{c} \left(H_{1},0.3715\right),\;\left(H_{2},0.0937\right),\;\left(H_{3},\;0.0460\right),\\ \;\left(H_{4},0.2394\right),\left(H_{5},0.2494\right) \end{array}\right\}$
	Flexible departure interval $\left(e_{32}\right)$	$\left\{\begin{array}{c} \left(H_{1},0.4550\right),\;\left(H_{2},0.1780\right),\;\left(H_{3},0.2383\right),\;\\ \left(H_{4},0.0175\right),\left(H_{5},0.1112\right) \end{array}\right\}$
Comfort $\left(S_{4}\right)$	Timely information ($e_{41}$)	$\left\{\begin{array}{c} \left(H_{1},0.3660\right),\;\left(H_{2},0.2494\right),\;\left(H_{3},0.1167\right),\;\\ \left(H_{4},0.2494\right),\left(H_{5},0.0186\right) \end{array}\right\}$
	Facility acquisition $\left(e_{42}\right)$	$\left\{\begin{array}{c} \left(H_{1},0.3710\right),\;\left(H_{2},0.3753\right),\;\left(H_{3},0.0542\right),\;\\ \left(H_{4},0.1904\right),\left(H_{5},0.0092\right) \end{array}\right\}$
Economy $\left(S_{5}\right)$	Convenient ticket purchase $\left(e_{51}\right)$	$\left\{\begin{array}{c} \left(H_{1},0.4356\right),\;\left(H_{2},0.1891\right),\;\left(H_{3},0.2518\right),\;\\ \left(H_{4},0.0819\right),\left(H_{5},0.0416\right) \end{array}\right\}$
	Cheap far e $\left(e_{52}\right)$	$\left\{\begin{array}{c} \left(H_{1},0.4871\right),\;\left(H_{2},0.0129\right),\;\left(H_{3},0.0743\right),\;\\ \left(H_{4},0.2427\right),\left(H_{5},0.1830\right) \end{array}\right\}$
	various tickets $\left(e_{53}\right)$	$\left\{\begin{array}{c} \left(H_{1},0.2524\right),\;\left(H_{2},0.4837\right),\;\left(H_{3},0.0128\right),\;\\ \left(H_{4},0.1134\right),\left(H_{5},0.1277\right) \end{array}\right\}$

**Table 5 entropy-22-00750-t005:** The combined assessment sets SS($S_{m}$) of 5 state indices.

State Indices	Assessment sets SS $\left(S_{m}\right)$
Safety $\left(S_{1}\right)$	$\left\{\left(H_{1},0.4231\right),\;\left(H_{2},0.2171\right),\;\left(H_{3},0.1793\;\right),\;\left(H_{4},0.1141\right),\left(H_{5},\;0.0665\right)\right\}$
Reliability $\left(S_{2}\right)$	$\left\{\left(H_{1},0.4433\right),\;\left(H_{2},0.1350\right),\;\left(H_{3},0.0178\right),\;\left(H_{4},0.2993\right),\left(H_{5},0.1046\right)\right\}$
Convenience $\left(S_{3}\right)$	$\left\{\left(H_{1},\;0.4467\right),\;\left(H_{2},0.1294\right),\;\left(H_{3},0.1325\right),\;\left(H_{4},0.1172\right),\left(H_{5},0.1742\right)\right\}$
Comfort $\left(S_{4}\right)$	$\left\{\left(H_{1},0.3822\right),\;\left(H_{2},0.3145\right),\;\left(H_{3},0.0776\right),\;\left(H_{4},0.2134\right),\left(H_{5},0.0123\right)\right\}$
Economy $\left(S_{5}\right)$	$\left\{\left(H_{1},0.3520\right),\;\left(H_{2},0.3456\right),\;\left(H_{3},0.1261\right),\;\left(H_{4},0.0947\right),\left(H_{5},0.0815\right)\right\}$

**Table 6 entropy-22-00750-t006:** The importance rakings of Rs.

Response Indices	Symbol	$E\left(Y_{j}\right)$	Ranking
advancement of design standards	$R_{1}$	0.6960	1
Popularization of informatization	$R_{2}$	0.5335	6
completeness of operational rules	$R_{3}$	0.5847	3
rationality of passenger flow control	$R_{4}$	0.5657	5
standardization of management operation	$R_{5}$	0.5769	4
Coordination of emergency linkage	$R_{6}$	0.4241	9
Rationality of service process	$R_{7}$	0.5259	7
Suitability of training standards	$R_{8}$	0.4309	8
reliability of subway facilities	$R_{9}$	0.6358	2
Innovation of the pricing mechanism	$R_{10}$	0.3139	11
Order of ticket card management	$R_{11}$	0.3450	10

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
