# Peer review of "Research on Evaluating the Sustainable Operation of Rail Transit System Based on QFD and Fuzzy Clustering"

_entropy, 2020, doi:10.3390/e22070750_

Round 1

Reviewer 1 Report

The paper presents a solid contribution to the application of fuzzy clustering analyses and therefore should state this more visible. The title of the paper is misleading due to "Research on Evaluating the..." and the acronyms; it should be something like "A fuzzy clustering approach for evaluating..." (or even fuzzy cluster analysis for...), since this is the main contribution of the paper.

Besides this, the authors should rework the whole language of the paper. Even the abstract is full of missing words and followingly incomplete sentences. Several parts of the paper need even more formulations as in subsection 3.3, since it is hard to follow the structure and sequence of applying the 3-step evaluation method. Additionally, I suggest to introduce a short introduction to section 3 immidiately after the section heading. The authors might consider to introduce the component of the 3 steps in a more overviewable manner as again counting the components beginning with 1...; this is confusing, since those are within only one subsection. I would suggest to have diagrams or switching to capital letters beginning with A. The first option would extend the diagram in figure 2.

Another point, that would highly benefit in representation is the introduction; here the authors may rewrite it to a more concisely-written section. Formulations like "As above discussed" and "What’s more" are not very suitable for the introduction. A rewriting aiming longer sentences with concise content and a lower number of sencences in the introduction would highly benefit the, besides the language, valuable overview on the topic with an well-selected references exceeding the number of 60.

In my opition, reworking the mentioned points would highly improve the paper.

Reviewer 2 Report

This manuscript is dedicated to the sustainable operation of rail transit systems by PSR and QFD, which is an important condition for urbanization.

Advantages of this work:
-Clear and consistent presentation.
-An original look at the problem and solution.

Suggestions for improvement:
-The introduction is busy, maybe it makes sense to highlight a separate section contains "Literature review".
-Manuscript sections regarding "Materials and Methods" contains many references and information that could be placed in other sections, such as "Introduction", "Literature review", "Preliminaries" or "Discussion".
-Directly specifying numbers (13, 5) in formulas is not very good practice (i.e., formulas 1, 3, and other).
-Conclusions provide a brief description of the manuscript but do not reflect the conclusions themselves. And there is no clearly outlined future work.
-Figures 1, 2 are smooth.
-Labels in Fig 2 are turned to the right but the left turn is expected.
-The References consists mainly of works by local authors.
-There is an inaccurate use of articles, for example, line #40 "A integrated..."

English proofreading is required.

The manuscript is written clearly in good language. All the results archived are commented enough.
The article can be accepted after minor revision.
